# Miniaturized Photo-Ionization Fourier Deconvolution Ion Mobility Spectrometer for the Detection of Volatile Organic Compounds

**DOI:** 10.3390/s22155468

**Published:** 2022-07-22

**Authors:** Binwang Yang, Jianna Yu, Wen Liu, Guoxing Jing, Wenshan Li, Wenjie Liu

**Affiliations:** College of Chemical Engineering, Xiangtan University, Xiangtan 411105, China; yanginu@smail.xtu.edu.cn (B.Y.); yujianna2000@sina.com (J.Y.); helenliu1990@163.com (W.L.); xing810810@163.com (G.J.); liwenshan@xtu.edu.cn (W.L.)

**Keywords:** multiplexing, Fourier deconvolution, miniaturized drift tube, photoionization, ion mobility spectrometer

## Abstract

Because of its simplicity, reliability, and sensitivity, the drift tube ion mobility spectrometer (IMS) has been recognized as the equipment of choice for the on-site monitoring and identification of volatile organic compounds (VOCs). However, the performance of handheld IMS is often limited by the size, weight, and drift voltage, which heavily determine the sensitivity and resolving power that is crucial for the detection and identification of VOCs. In this work, we present a low-cost, miniaturized drift tube ion mobility spectrometer incorporated with a miniaturized UV ionization lamp and a relatively low drift voltage. The sensitivity and resolving power are boosted with the implementation of Fourier deconvolution multiplexing compared to the conventional signal averaging data acquisition method. The drift tube provides a high resolving power of up to 52 at a drift length of 41 mm, 10 mm ID dimensions, and a drift voltage of 1.57 kV. Acetone, benzene, dimethyl methyl phosphonate, methyl salicylate, and acetic acid were evaluated in the developed spectrometer and showed satisfactory performance.

## 1. Introduction

Drift tube ion mobility spectrometry (IMS) is a rapid gas-phase detection technique, in which various ions are driven by an electric field and in constant collisions with counter-current drift gas to achieve atmospheric pressure separation. Drift tube ion mobility spectrometry plays an important role in explosives, chemical warfare agents, and drug detection [1,2,3,4] because of its advantages of great stability, sensitivity, fast response, low cost, and good portability. With the continuous development of IMS, solid, low-cost, and lightweight portable or handheld drift tubes have shown potential use in environmental monitoring [5], food safety [6], and drug analysis [7,8]. Therefore, the development of miniaturized high-performance IMS is of great significance.

The ionization source of the conventional miniaturized drift tube IMS is usually radioactive [1,9,10,11,12], such as ^63^Ni, ^3^H, and ^241^Am. A common feature of radioactive ionization sources is that they emit high-energy particles whose ionization energy far exceeds the ionization energy of various analyte molecules and all background air components, resulting in complex ion-molecule reactions in the drift tube ionization region [13,14], low ionization selectivity, and a complex IMS spectrum. Another concern about the radioactive source is the regulatory restrictions relating to radiation safety and the problems with storage and transport. Therefore, non-radioactive ionization sources are much more desired, such as an ultra-violet photo-ionization source, a corona-discharge ionization source, and an electrospray ionization source [15,16,17,18,19,20,21]. Of them, the ultra-violet photo-ionization source, is favored for its simplicity, size, and the growing concern about the health and environmental effect of radioactive sources.

The sensitivity of the photo-ionization IMS system is heavily affected by the ionization capability, which largely depends on the size of the ultra-violet lamp, generally a 10.6 eV Kr lamp. Though miniaturized lamps are commercially available, they are seldom deployed in miniaturized drift tube IMS sensing systems. Other major factors that affect the performance of IMS are the resolving power *R_p_* and the signal-to-noise ratio (SNR). The resolving power is defined as the quotient of the drift time **t**_d_** and the full width at half the maximum *W_FWHM_* of a peak in the spectrum, as in the following equation:(1)$$R_{p}=\frac{t_{d}}{w_{FWHM}}$$

The SNR is a measure for the limits of detection (LOD) and thus the minimum concentration of a sample component required for detection. The higher the SNR, the better the LOD.

The conventional IMS mostly adopts the single pulse signal averaging method. The ion mobility spectrum is initiated by opening the ion gate to introduce a packet of ions into the drift region for separation. Short ion gate opening widths corresponds to narrow initial ion packet widths and hence increase the resolving power. However, at a reduced pulse width, the number of introduced ions is also decreased and hence lowers the signal intensity consequently. The pulse width should be prolonged to increase the SNR but lower the resolving power. To resolve the problem between SNR and resolving power, an effective approach is to use multiplexing techniques, such as Fourier transform [22,23,24,25,26], Hadamard transform [18,27,28,29], matched filtering (cross-correlation) [30], and Fourier deconvolution [31]. Fourier deconvolution IMS is use modulation of the ion gate with linear chirp signals, in which the duty cycle for ion injection is improved to 50%, with high SNR and resolving power without hardware modifications [31], greatly improving the performance of IMS.

In this study, we demonstrate a low-cost, miniaturized drift tube incorporated with a miniaturized UV ionization source and a relatively low drift voltage for handheld and portable devices in the analysis of volatile organic compounds (VOCs) for air quality monitoring, toxic industrial chemicals detection, and chemical warfare agent detection. To boost the resolving power and signal-to-noise ratio, the Fourier deconvolution algorithm is integrated into the data acquisition and spectrum reconstruction process.

## 2. Experimental Section

### 2.1. Chemicals and Reagents

Acetone (AR) was purchased from Hunan Huihong Reagent Company, Ltd. (Hunan, China). Benzene (AR) was purchased from Sinopharm Chemical Reagent Company, Ltd. (Shanghai, China). Dimethyl methyl phosphonate (DMMP) (AR) was purchased from RHAWN (Shanghai, China). Methyl salicylate and acetic acid were purchased from Aladdin (Shanghai, China).

### 2.2. Miniaturized Photo-Ionization Ion Mobility Spectrometer Drift Tube

The miniaturized drift tube consisted of five main parts: ionization source, reaction region, ion gate, drift region, and a Faraday plate as the ion detector. The drift tube was made with alternatively stacked polyether ether ketone (PEEK) insulation rings and stainless-steel electrodes. The PEEK rings and stainless-steel rings were sealed with one component of epoxy heat-curing adhesive (DELO MONOPOX HT2860, Munich, Germany). To afford enough bonding between PEEK and stainless steel, both materials were surface treated with low-temperature plasma prior to the bonding process. The thicknesses of the PEEK insulation ring in the drift region and reaction region were 2.4 mm and 3.3 mm, respectively. The thickness of the stainless-steel electrode ring was 0.5 mm. A Tyndall–Powell gate (TPG) was installed between the reaction region and the drift region. The electric field in the drift tube was provided by a series of stainless-steel electrodes, which were inserted into the printed circuit board (PCB) and connected with a 1M chip resistor to the drift voltage.

A commercial radio-frequency excited ultra-violet (RF UV) lamp with photon energy of 10.6 eV (PKR106-6-14, Heraeus, Chesterfield, UK) was used in the photo-ionization (PI) IMS drift tube as the ionization source, which was axis-mounted in front of the drift tube. The on-axis design of the ionization UV lamp shows better limits of detection than other designs [32]. Under the atmosphere conditions, the penetration depth of the UV photons was shortened to around 1.5 cm due to oxygen absorption; thus, the length of the ionization region was determined to be 1.5 cm. The UV lamp was 6 mm in diameter (output window) and 14 mm in length, operated with a 13 MHz power supply (C210, Heraeus, Chesterfield, UK).

The electric field in the UV-IMS drift region was set as 383 V·cm^−1^. The TPG is operated with a closure voltage of ±25 V, and the gate opening pulse width is 100 μs in normal pulsed mode using a homemade TPG gate controller. A stainless-steel grid is located in front of the reaction region as the repulsion electrode. In the detection region, a stainless-steel grid was placed in front of a Faraday plate at a distance of 1 mm. A high voltage capacitor (2 kV, 100 pF) is connected between the reference voltage and round to reduce the impact from the ion gate switching signal, and another Polypropylene capacitor (400 V, 1 μF) is connected between the aperture grid and ground. The received ion signal on the Faraday plate is amplified by a custom-made ADA4530-1-based two-stage pre-amplifier with 5 × 10^8^ before transmission to data acquisition system (DAQ) equipment. The drift tube temperature was kept at 25 °C. A unidirectional flow scheme was adopted in the PI-IMS [17,33,34]. The sample was introduced into the photoionization chamber by a carrier gas, and the drift gas was brought into the drift tube from the back of the Faraday plate to keep the drift tube free from contamination. Dry purified air was purified and compressed by an Air Generator (QPA-2LP, Quanpu, Shanghai, China), a silica gel trap, and an activated carbon trap, and a 13× molecular sieves trap was used as carrier gas and drift gas. The moisture level of the purified compressed air was kept below 1 ppm_v_. The flow rates for carrier and drift gases were 30 mL/min and 100 mL/min, respectively, and adjusted by two mass flow rate controllers (LZB-3WB, Shuanghuan, Changzhou, China). The atmospheric pressure was 101.2 kPa during the experiment conducted. A photograph illustration of the home-built PI-IMS is presented in Figure 1. The drift tube is plugged into a PCB motherboard with surface-mounted resistors (SMD resistors) consisting of a linear gradient field; the external dimension of the drift tube is 20 mm × 20 mm, and the inner diameter is 10 mm. The length of the drift region is 41 mm. The main parameters of the PI-IMS are shown in Table 1.

### 2.3. Fourier Deconvolution Modulation Sequence Generation and Data Acquisition

The Fourier deconvolution ion gate control function is a linear increasing frequency square wave sequence. The Signum function transforms the linear increasing chirp signal starting from 0 Hz into an ion gate opening and closing modulation sequence signal *m*(*t*) for ion gate driving [31]:(2)$$m\left(t\right)=sgn(\cos\left(c_{0}+\frac{\pi K}{Z}t^{2}\right))$$
where *m* is the produced modulation sequence signal, *K* is the modulation chirp frequency, *Z* is the modulation cycle, *sgn* is the Signum function, *t* is the data acquisition time, and *c*_0_ is the original phase.

A homemade LabVIEW-based (National Instruments, Houston, TX, USA) program was used for ion signal data acquisition and ion gate control modulation sequence signal generation. A Peripheral Component Interconnect Data Acquisition (PCI DAQ) board (PCI-6251, National Instruments, Houston, TX, USA) was employed to generate modulation sequence and (analog-to-digital converter) ADC sampling. The maximum sampling rate of the PCI DAQ board was 500 k 16-byte.

### 2.4. Sample Preparation

The gaseous standards of acetone, benzene, dimethyl methyl phosphonate, methyl salicylate, and acetic acid were prepared with an exponential dilution method. The volume of the dilution glass flask was 500 mL. A syringe was used to inject gas analytes through a stainless-steel sleeve tee with a diaphragm; the diluted sample was then conducted to the IMS carrier gas inlet. The calculation formula of sample concentration (*y*) is:$$y=y_{0}e^{-\frac{Lt}{A}}$$
where *y*_0_ is the original concentration of the gas sample after complete diffusion in the glass flask, *A* is the volume of the glass flask used for dilution, *t* is the dilution time of the gas sample, and *L* is the flow of purified gas into the dilution flask.

## 3. Results and Discussion

### 3.1. Reduction in Ion Gate Switching Impact on Ion Mobility Spectra

Most drift tube ion mobility spectrometers use Faraday plates as ion detectors; the induced signal by the ion gate switching electromagnetic pulse is generally significant for a miniaturized drift tube. For the traditional signal-averaging method, the induced signal and real ion mobility peaks are located in different drift times, and there are no steps needed to deal with this induced signal at the beginning of ion mobility spectra. However, for multiplexing IMS experiments, the induced signal is superimposed on the real ion signals. Although the ion signal and induced signal can be decomposed by the deconvolution algorithm, the use of low-pass filter treatment before deconvolution significantly deteriorates the deconvolution ion mobility spectra because the induced signal is much stronger than the real ion signal and causes serious baseline distortion. To address this problem, further steps are needed to remove this induced signal from real ion signals. Instead of shielding this induced signal by gate switching, a simple mathematic method is implemented to the standard Fourier deconvolution process as follows:

The time-domain ion mobility signal with an extremely narrow gate pulse width is defined as *r*(*t*). The ion gate is controlled by modulated gated sequence signal multiplexing, then the multiplexed ion signal obtained in the detector is the convolution output of modulated gated sequence signal *m*(*t*) and time-domain ion mobility signal *r*(*t*)*:*(3)$$n\left(t\right)=m\left(t\right)*r\left(t\right)=(\int_{-\infty }^{\infty }m\left(t\right)r\left(x-t\right)d\tau )$$
where *n*(*t*) is the detected multiplexed ion signal. According to the convolution theorem, the Fourier transform of function convolution is the product of function Fourier transform, so we know
(4)$$FF\left(n\left(t\right)\right)=FF\left(m\left(t\right)*r\left(t\right)\right)=FF\left(m\left(t\right)\right)FF\left(r\left(t\right)\right)$$

The time-domain ion mobility signal obtained from multiplexed ion signal is actually to solve *r*(*t*) with the known modulation gated sequence function *m*(*t*) and the detected *n*(*t*)*:*(5)$$r\left(t\right)=IF(\frac{F\left(n\left(t\right)\right)}{F\left(m\left(t\right)\right)})$$
where *F* and *IF* denote Fourier transform and inverse Fourier transform, respectively.

To avoid the serious ring effect caused by the subsequent Fourier low-pass filter, the opening and closing of the ion-gate-induced pulse signal in the direct deconvoluted time-domain ion mobility signal *r*(*t*) were replaced with zeros and then subjected to a Fourier low-pass filter to remove the high-frequency noise.

### 3.2. Reconstructed Process of Ion Mobility Spectra

To evaluate the effectiveness of the modified signal-processing method, a 1 s duration, 0–10 kHz modulation sequence (Figure 2a) was generated to control the ion gate, and the multiplexed ion current signal was synchronously sampled (Figure 2b). The data acquisition and ion gate modulation rates were both set to 200 ksps. To eliminate the ion gate switching induced pulse signal (Figure 2c), the first 0.25 ms of the time-domain ion mobility signal is zero filled. To filter out possible noise subjected to ion current, the time-domain ion mobility signal is then transformed into the frequency domain using a fast Fourier transform (FFT) directly and then low-pass filtered using the same frequency as the gate modulation frequency. Figure 2 demonstrates the multiplexed ion current of acetone ion, the time domain ion mobility signal, and the output ion mobility spectrum (Figure 2c–f, 210 ppm_V_ acetone).

### 3.3. The Influence of Various Modulation Chirp Frequencies, Data Acquisition Rate and ADC Sampling Rate to the Performance of Miniaturized IMS Spectra

To study the performance differences among various modulation chirp frequencies, we compared 8 kHz,10 kHz,12 kHz, 15 kHz, 20 kHz, and 25 kHz, each with a duration of 1 s (Figure 3). Each modulation chirp frequency was performed 20 times, and the results were averaged. From Figure 3, the signal-to-noise ratio decreases rapidly along the increase in modulation frequency, while the resolving powers are mainly unchanged when modulation frequency exceeds 10 kHz. For this reason, the optimal modulation frequency was decided at 10 kHz.

Another important parameter in the modulation process is the data acquisition period. When the ion mobility spectrometry peak is measured in seconds, the shorter data acquisition period is conducive to sampling. In the experiment, the final frequency of the modulation sequence was 10 kHz, and the data acquisition time was from 100 ms to 2000 ms (Figure 4). The experimental results show that the spectrum acquisition time of FD-IMS can be reduced to 200 ms without deteriorating the quality of the acquired spectrum, equivalent to 5 spectra per second. When the data acquisition time is between 200 ms and 2000 ms, the resolving power fluctuation is very small, which can be regarded as not affected by the modulation time, and the signal-to-noise ratios increase with the increase in data acquisition times.

The synchronous increase in the ADC sampling rate and ion gate modulation rate results in high signal-to-noise ratio and high resolving power at the expense of larger sets of data (Figure 5). From this point of view, 200 ksps is a compromise between data size and spectra performance, resulting in a signal-to-noise ratio of 2370 and resolving power of 45.3.

### 3.4. Analytical Performance of the Miniaturized Drift Tube

Using the optimized conditions, we compared the performance of miniaturized photo-ionization FD-IMS with the conventional signal average data acquisition method using benzene vapor. The modulation chirp sequence for FD-IMS was from 0 to 10 kHz, and the data acquisition time was 1 s. Figure 6a shows the ion mobility spectrum of 130 ppm_V_ benzene detected by the Fourier deconvolution method; the resolving power of the main peak of benzene (*t**_d_* = 5.9 ms) is 47, and the signal-to-noise ratio is 993.6. Figure 6b shows the ion mobility spectrum of 130 ppm_V_ benzene detected by the traditional signal average data acquisition method (DSA) using the same low-pass filter as FD-IMS, while the resolving power of the main peak of benzene is 38, and the signal-to-noise ratio is 140. The signal-to-noise ratio of the Fourier deconvolution method is 7.1 times higher than that of the traditional signal average data acquisition method.

In Figure 7a, the positive photoionization ion mobility spectrum of 100 ppm_V_ acetone is shown. In Figure 7b, the dopant-assisted negative photoionization ion mobility spectrum of 100 ppm_V_ Acetone dopant is shown. The resolving power of the positive reactant ion peak (acetone ion) at *t**_d_* = 5.86 ms is *Rp* = 46, while the resolving power of the negative reactant ion peak at *t**_d_* = 4.70 ms (CO_3_^−^ peak) and *t**_d_* = 5.00 ms (O_2_^−^ peak) are *Rp* = 52 and *Rp* = 45, respectively. The reduced mobilities of O_3_^−^ (H_2_O)_n_ and O_2_^−^ (H_2_O)_n_ are 2.18 cm^2^/(V·s) and 2.05 cm^2^/(V·s), respectively.

Under the photoionization conditions, we performed measurements on three different chemicals in clean, dry air. Dimethyl methyl phosphonate (DMMP) was characterized in positive ion mode, while methyl salicylate and acetic acid were characterized in the dopant-assisted negative ion mode, in which all voltage polarities are inverted. In Figure 8a, the positive ion mobility spectrum of 0.5 ppm_V_ DMMP with its monomer at *t**_d_* = 5.88 ms and dimer at *t**_d_* = 7.19 ms is shown. In the dopant-assisted negative mode 20 ppb_V_ methyl salicylate spectrum shown in Figure 8b, a double negative reactant ion peak and a single methyl salicylate peak appear, in which methyl salicylate at *t**_d_* = 7.06 ms. For a measuring time of 1 s, the detection limits of the DMMP monomer and dimer are about 0.41 ppb_V_ and 0.74 ppb_V_, and the achieved limits of detection are 1.40 ppb_V_ for methyl salicylate.

In Figure 9, a dopant-assisted negative ion mobility spectrum of 5 ppm_V_ acetic acid, with *t**_d_* = 5.65 ms, is shown. The detection limits the acetic acid are about 4.72 ppb_V_.

## 4. Conclusions

In this work, we present a low-cost, miniaturized drift tube ion mobility spectrometer incorporated with a miniaturized UV ionization source and a relatively low drift voltage. The drift tubes for PI-IMS consist of alternating layers of stainless-steel electrode sheets and PEEK insulator material, which are connected in a mechanically stable and gastight manner. The sensitivity and resolving power are boosted with the implementation of Fourier deconvolution ion mobility spectrometry compared to the conventional signal-averaging data acquisition method. The drift tube provides a high resolving power of up to 52 at a drift length of 41 mm, 10 mm ID dimensions, and a drift tube voltage of 1.57 kV. The limits of detection for one second of data acquisition are 0.41 ppb_V_ for dimethyl-methyl phosphonate monomer, 0.74 ppb_V_ for the dimer and 1.4 ppb_V_ for methyl salicylate.

## Figures and Tables

**Figure 1 sensors-22-05468-f001:**
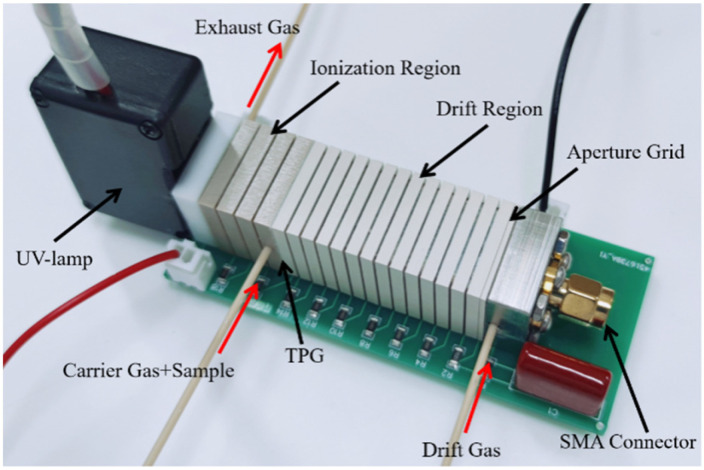
Photograph of the PI-IMS system.

**Figure 2 sensors-22-05468-f002:**
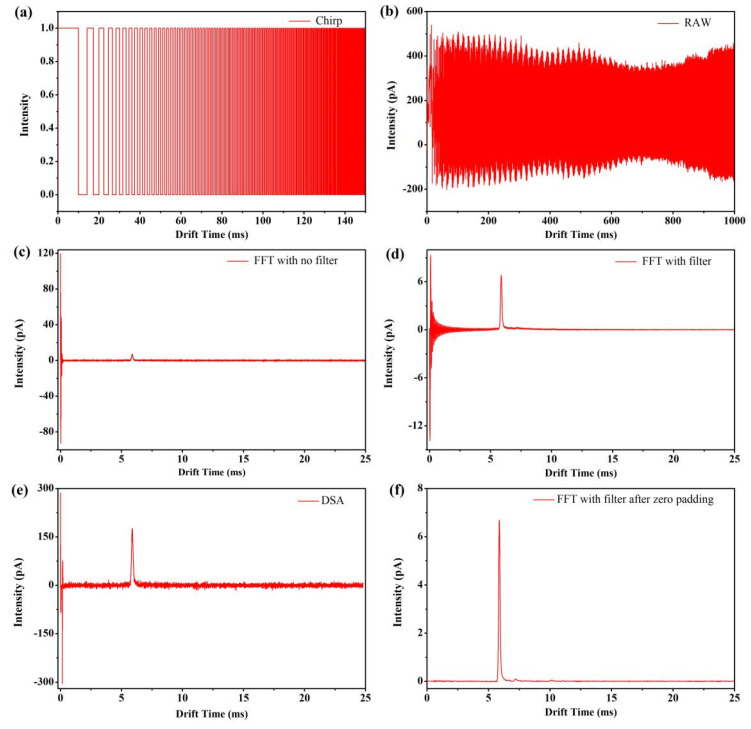
The modulation gate sequence of miniaturized Fourier deconvolution ion mobility spectrometer, multiplexed ion signal, and IMS spectra before and after eliminating baseline distortion. (**a**) Modulation gate sequence of FD-IMS (the first 150 ms is shown). (**b**) Detected multiplexed ion signal. (**c**) Primary ion mobility spectrum without low-pass filter before inverse Fourier transform. (**d**) Fourier deconvolution ion mobility spectrum of baseline distortion. (**e**) IMS spectrum of the signal-averaging method. (**f**) Fourier deconvolution ion mobility spectrum of eliminating baseline distortion.

**Figure 3 sensors-22-05468-f003:**
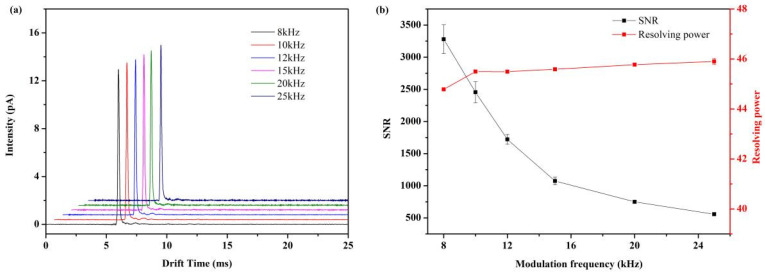
The influence of various modulation chirp frequencies on the performance of Fourier deconvolution IMS. (**a**) FD-IMS spectra of acetone (290 ppm) using 8000–25,000 Hz. (**b**) Signal-to-noise ratio and resolving power under various modulation chirp frequencies.

**Figure 4 sensors-22-05468-f004:**
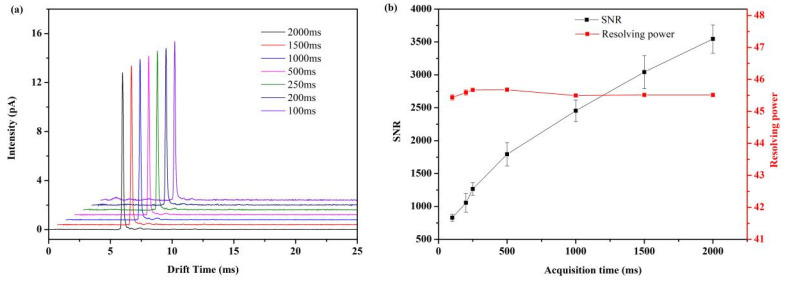
The effect of different data acquisition times to the signal-to-noise ratio and the resolving power of FD-IMS. (**a**) Acetone (290 ppm) IMS spectra with different data acquisition times. (**b**) Signal-to-noise ratio and resolving power under different data acquisition times.

**Figure 5 sensors-22-05468-f005:**
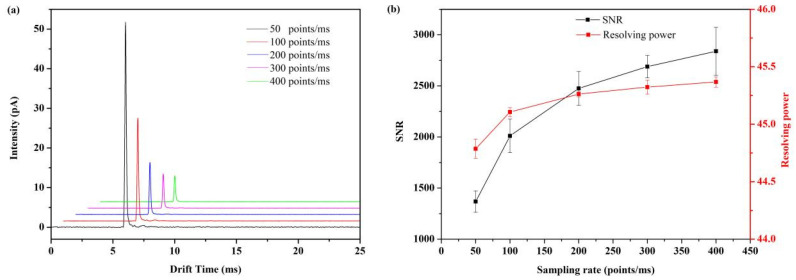
The influence of various ADC sampling rate to the performance of FD-IMS. (**a**) Acetone (290 ppm) IMS spectra with various ADC sampling rate. (**b**) Signal-to-noise ratio and resolving power under various ADC sampling rate.

**Figure 6 sensors-22-05468-f006:**
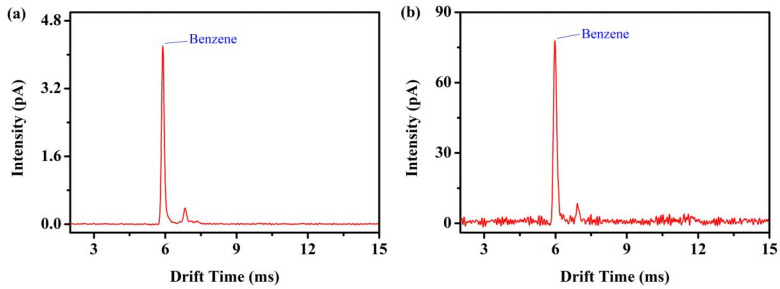
Ion mobility spectrum of benzene in positive photoionization (**a**) FD-IMS and (**b**) DSA-IMS.

**Figure 7 sensors-22-05468-f007:**
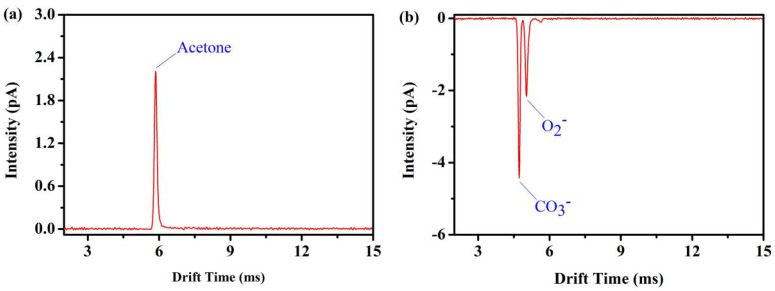
(**a**) Positive reactant ion peak (acetone) and (**b**) negative double reactant peaks (CO_3_^−^ and O_2_^−^) employing photoionization source FD ion mobility spectrometer, both in purified, dry air.

**Figure 8 sensors-22-05468-f008:**
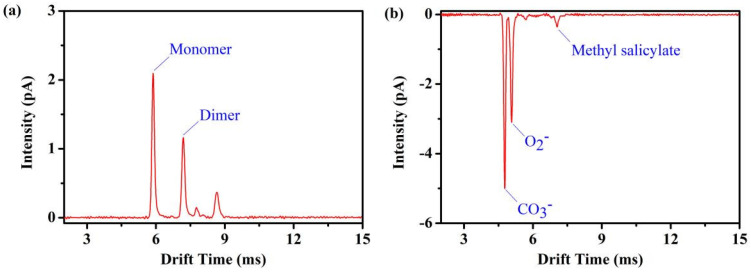
(**a**) Positive ion mobility spectrum of DMMP and (**b**) negative ion mobility spectrum of methyl salicylate obtained by the photoionization source FD ion mobility spectrometer.

**Figure 9 sensors-22-05468-f009:**
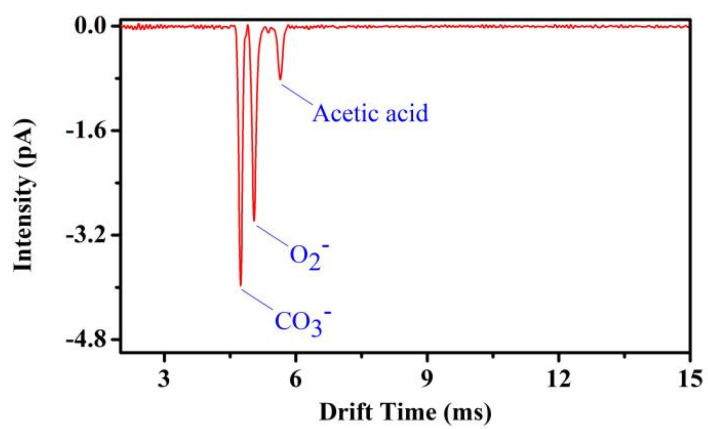
Ion mobility spectrum of acetic acid obtained by the dopant-assisted negative photoionization source FD ion mobility spectrometer.

**Table 1 sensors-22-05468-t001:** Main parameters of the PI-IMS system.

Parameter	PI-IMS
Length of Drift Region	41 mm
The inner diameter of the drift tube	10 mm
Electrical Field Strength	383 V/cm
Gate Voltage Differences	±25 V
Gate Pulse Frequency	0~10 kHz
Gain of Amplifier	5 × 10^8^ V/A
Ionization Source	10.6 eV UV Lamp
Drift Gas Flow Rate	100 mL min^−1^
Carrier Gas Flow Rate	30 mL min^−1^
Temperature	25 °C
Pressure	103.0 kPa

## Data Availability

The data presented in this work are available in the article and Appendix A.

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
