# Peer review of "Miniaturized Photo-Ionization Fourier Deconvolution Ion Mobility Spectrometer for the Detection of Volatile Organic Compounds"

_sensors, 2022, doi:10.3390/s22155468_

Round 1

Reviewer 1 Report

In this paper, the author designs a Fourier deconvolution ion mobility spectrometer, which can obtain detection results with high sensitivity and resolution. The proposed method has reference significance. However, I have some remarks:

1. Please give the concentration of the sample gas used in all experiments.

2. In the spectrum of benzene shown in Figure 6, is the DSA-IMS detection result using a low-pass filter? If not, what is the result of using low-pass filtering?

3. In lines 278-280, "For a measuring time of 1 s, the detection limits the DMMP monomer, dimer and trimer are about 0.41 ppbv and 0.74 ppbv...", three substances are listed in the paper, but only two detection limits are given.

Reviewer 2 Report

In this manuscript, the authors proposed using Fourier deconvolution ion mobility spectrometry incorporated with a miniaturized drift tube and miniaturized photo-ionization lamp for the detection of volatile organic compounds. Various parameters including multiplexing sequence, modulation time, and DAQ sampling rate were investigated. Considering the length of the drift tube, the relatively low drift voltage, and a miniaturized photo-ionization lamp, the developing instrument shows impressive performance such as resolving power and signal-to-noise ratio. The idea is interesting and there is clearly an improvement to this approach. Considering the increasing interest and importance of VOC detection, I would like to propose an acceptance after some minor revisions.

(1) When comparing the performance of different methods, the authors should provide accurate resolving power and signal-to-noise ratio in Figure 6.

(2) The authors should provide the unit of intensities in figure 2a to figure 2f. Why the intensity is decreased by almost 20 times when the signal-to-noise ratio is significantly improved.

(3) In the introduction section, the authors neglect to cite the early works using photo-ionization sources for the detection of VOCs.

(4) There are still some typos and grammar mistakes that should be corrected, such as P7, line 19, ‘without deteriorate’ should be ‘without deteriorating’

(5)The section of 3.2 seems missing.

Reviewer 3 Report

The idea of this work  is good and realization of it is also good. Before making decision concerning publication of the paper some remarks should be taken into account.

L. 83. Instead of Polyether should be polyether

L. 91. PCB acronym should be clarified

L. 93. What RF means?

L. 94. What PI means?

L. 96. LODs should be removed

L. 101. What TPG means

L. !07. What CBB means?

L. 110. What DAQ means?

L. 114-116. This sentence should be improved. Air generator is not purifier.

L. 121. What SMD means?

Table 1. After „carrier gas”  „flow rate” should be added

L. 139. What does it mean that program from National Instruments was home-made?

L. 141. What is PCI DAQ?

L. 142. What ADC means?

L. 150. Instead of y should be c

L. 162-165. This sentence is unintelligible

L. 190. What FD means?

L. 203. What FFT means?

L. 206. Fig. 2c-2e probably are also peaks of acetone

Figure 3. The lines in 3b should be denoted

L. 212. What is resolving power? It should be defined. How it is calculated?

L. 225. What about the chromatographic peak?

L. 239. What 200K means?

L. 243. What DSA means?

L. 248. Instead of benzyne should be benzene

L. 253-254. From the fig. 6 it is not visible that SNR for Fourier deconvolution is 7.1 times higher than for traditional method

The use of dopant should be clarified. And was it good choice? The drift time of it (5.86) is almost exactly the same as DMMP (5.88).
